# Versatile Applications of Cyanobacteria in Biotechnology

**DOI:** 10.3390/microorganisms10122318

**Published:** 2022-11-23

**Authors:** Ewa Żymańczyk-Duda, Sunday Ocholi Samson, Małgorzata Brzezińska-Rodak, Magdalena Klimek-Ochab

**Affiliations:** Faculty of Chemistry, Department of Biochemistry, Molecular Biology, and Biotechnology, Wroclaw University of Science and Technology, Wybrzeże Wyspiańskiego 29, 50-370 Wrocław, Poland

**Keywords:** cyanobacteria, bioactive compounds, bioremediation, photobiocatalysis

## Abstract

Cyanobacteria are blue-green Gram-negative and photosynthetic bacteria which are seen as one of the most morphologically numerous groups of prokaryotes. Because of their ability to fix gaseous nitrogen and carbon dioxide to organic materials, they are known to play important roles in the universal nutrient cycle. Cyanobacteria has emerged as one of the promising resources to combat the issues of global warming, disease outbreaks, nutrition insecurity, energy crises as well as persistent daily human population increases. Cyanobacteria possess significant levels of macro and micronutrient substances which facilitate the versatile popularity to be utilized as human food and protein supplements in many countries such as Asia. Cyanobacteria has been employed as a complementary dietary constituent of feed for poultry and as vitamin and protein supplement in aquatic lives. They are effectively used to deal with numerous tasks in various fields of biotechnology, such as agricultural (including aquaculture), industrial (food and dairy products), environmental (pollution control), biofuel (bioenergy) and pharmaceutical biotechnology (such as antimicrobial, anti-inflammatory, immunosuppressant, anticoagulant and antitumor); recently, the growing interest of applying them as biocatalysts has been observed as well. Cyanobacteria are known to generate a numerous variety of bioactive compounds. However, the versatile potential applications of cyanobacteria in biotechnology could be their significant growth rate and survival in severe environmental conditions due to their distinct and unique metabolic pathways as well as active defensive mechanisms. In this review, we elaborated on the versatile cyanobacteria applications in different areas of biotechnology. We also emphasized the factors that could impede the implementation to cyanobacteria applications in biotechnology and the execution of strategies to enhance their effective applications.

## 1. Introduction

Cyanobacteria are blue-green algae that generate biomasses via the conversion of carbon dioxide using solar energy [1]. Cyanobacteria have a wide variety of colors such as pink, red, yellow, green and brown [2], and they exist in diverse ecosystems such as in rock surfaces, oceans, soil and freshwater [3]. It is important to know that cyanobacteria are the first universal oxygen-producing photosynthetic microbes and for many billions of years, they have contributed to the universe’s oxygen generation [4]. Cyanobacteria are microscopic organisms; however, they could be observed via unaided eyes in the form of blooms or colonies [5,6]. The potential of cyanobacteria to adapt to several environmental conditions, ability to generate oxygen and simple genomes (enabling easy manipulations), as sources of bioactive compounds, and their fast growth rate have prompted many researchers to perform thorough investigations on cyanobacteria as supplements in human food, animal feed, in industry and in medicine [7,8]. However, alterations in climate change, global warming, nutrients availability and biotic factors determine the rapid growth rate of cyanobacteria [9,10,11]. 

Cyanobacteria are abundant sources of bioactive compounds. Their secondary metabolites are rich sources of vitamins, enzymes and toxins which are crucial in many biotechnological industries [7], for instance, in the production of bioplastics from cyanobacteria’s polyhydroxy-alkanoates (PHA) [12,13]. PHA accumulates intracellularly in many cyanobacteria species, which can be utilized in bioplastics production with properties such as polyethylene and polypropylene [12]. A wide variety cyanotoxins have been generated from cyanobacteria [14]. These bioactive compounds among others include saxitoxins, microcystins, and anatoxins [15]. This review emerged to broadly elaborate the important roles of cyanobacteria in biotechnology.

## 2. Cyanobacterial Metabolites

Cyanobacteria exhibit metabolic processes such carotenogenesis and photosynthesis, which generate highly valued primary and secondary metabolites. Primary metabolites are those metabolites that are strictly involved in processes such as reproduction, growth and cell division, which are referred to as developmental processes [15]. Primary metabolites include antioxidants, primary proteins and lipids [16]. These metabolites could be re-engineered to produce important biotechnological products such as dyes, biofertilizers, food supplements and bioplastics [17,18,19,20]. However, secondary metabolites are not used by cyanobacterial cells and are not directly engaged in their normal growth, reproduction or development [21]. Secondary metabolites are typically distinctive to specific organisms and are not regularly available in all environmental conditions. Hence, they are produced for defensive purposes [22]. Secondary metabolites from cyanobacteria include toxins, phenolic compounds, essential oils, alkaloids, steroids and antibiotics [23,24,25,26,27,28,29]. 

Previous epidemiological research revealed that cyanobacteria have a quite simple genetic material [30], enabling easy manipulations and modifications for the discovery of new important metabolites. These metabolites could be produced naturally from cyanobacteria via their responses to environmental stresses such as alteration of light intensity and deficiencies in micro and macro nutrients, which could alter their photosynthetic cellular metabolism [31]. 

Chen et al. [32] elucidated the manipulation of cyanobacteria growth conditions by comparing two phases. In the first phase, the cyanobacteria were allowed to grow in suitable and favorable conditions, whereas in the second phase, cyanobacteria were allowed to grow in unfavorable environmental conditions such as nitrogen deficiency, elevated intensity of light and deficiency of nutrients. The results revealed significant levels of secondary metabolites such as essential oils, starch granules, lipids and biopolymers from the second phase when compared to the first stage [32,33]. Rachana et al. [34] uncovered the exploitation of the cyanobacteria metabolic pathway (Methylerythritol 4-phosphate, MEP) to generate valuable secondary metabolites such as chlorophyll, fatty acids, and hormones. These products possess antimicrobial, antibiotic and anticarcinogenic properties that facilitate their active roles in pharmaceutical industries [34,35]. 

### 2.1. Phenolic Acids

Phenolic acids are aromatic rings consisting of one or more hydroxyl groups and one carboxyl group. Phenolic acids produced by cyanobacteria are essential for the defense against oxidative destruction that could result from hydroxyl radicals and the reactive oxygen species [36]. The buildup of phenolic acids in cyanobacteria ensures the adaptability and tolerance of cyanobacteria against several environmental stresses such as interaction with heavy metals and UV light [37,38], which could result in the deposition of free radicals in cells, damage to deoxyribose and chemical destruction of DNA [39].

As an example, Singh et al. [40] investigated how phenolic acids produced by cyanobacteria played an important role in the rummaging of free radicals and ensuring tolerance and adaptability when the cultures of various cyanobacteria species (*Oscillatoria acuta*, *Plectonema boryanum*, *Anabaena doliolum* and *Haplosiphon intricatus*) were subjected to a high sodium chloride concentrations (considered as unfavorable condition). The experimental results revealed significant accumulation of phenolic acids including vanillic acid, gallic acid, chlorogenic acid, caffeic acid and ferulic acid [40]. In another instance, Tanutcha et al. [41] revealed that the treatment of the cyanobacteria specie, *Halothece* sp. PCC7418 cells with temperature shock allowed for the robust production of phenolic compounds and phycobiliproteins. The quantification of phenolic compounds and phycobiliproteins in aqueous extracts showed that the amounts of the metabolites were regulated by the hot and cold temperature shocks [41].

Phenolic acids produced by cyanobacteria with great scavenging of free radicals and detoxification of reactive oxygen species (ROS) facilitated their relevance as therapeutic agents. For instance, anti-winkle and skin whitening facilitated by ferulic acids [42], anti-aging effect facilitated by gallic acids [43], effective heart failure recovery facilitated by syringic, gentisic and gallic acids [44], as well as antimicrobial effects of other phenolic compounds were generated by cyanobacteria [45,46,47]. 

### 2.2. Vitamins

Vitamins participate in many metabolic pathways, acting as coenzymes. They also engage in diverse processes such as antioxidants, controlling and regulating cell functioning and growth of tissues [48]. There are various categories of vitamins utilized by living organisms. These include lipid-soluble vitamins (vitamin A, D, E, and K) and water-soluble vitamins (vitamin B1, B2, B3, B5, B6, B7, B9, B12 and C). Cyanobacteria produce vitamin A, B, C, E and K in response to environmental stress, as elucidated by Helliwell et al. [49] and Asensi-Fabado et al. [50]. Deficiency of vitamins in humans is a significant menace globally and necessitates immediate attention. In the notion for bioactive metabolites generation, cyanobacteria denote one of the most capable potentials with many biotechnological applications and for allowing the advancement of an eco-sustainable creation of natural bioactive metabolites. It is important to know that not all vitamins are generated by all plants and some vitamin (B, D and K) are scarce in many plants [48,50]. Hence, cyanobacteria have been demonstrated to generate these vitamins that are scarce in several vascular plants [48]. Individuals acquire significant number of vitamins via the consumption of fruits, vegetables and dairy products such meat, fish and eggs. However, an individual with some vitamins’ deficiency could be treated with artificial or synthetic vitamin, through oral or intramuscular delivery of vitamins: B_12_ (oral and intramuscular), D1 (oral ergocalciferol), C (oral) and K1 (intramuscular) [51,52,53,54,55].

Cyanobacterial products are fascinating and have gained increasing attention because of their high contents in vitamins, minerals, essential amino acids and protein [56]. As an example, *Spirulina* products are extensively promoted due to their high contents of vitamin B_12_, high contents of protein (>60%) and additional micronutrients [56]. *Spirulina* is known to be an important vitamin B_12_ source for vegans and vegetarians, as chemical analyses showed that vitamin B_12_ dry weight levels range from 127 to 244 μg/100 g [57]. The range could be rendered into a day-to-day *Spirulina* intake of 1.6–3.2 g to reach the satisfactory consumption of 4 μg/day for cobalamin (vitamin B_12_) according to the EFSA [58]. However, the suppliers of *Spirulina* suggested a regular intake of 3–9 g/day would be efficient to enhance vitamin B_12_ requirements [59]. So far, *Arthrospira* sp. (*Spirulina*) has been made accessible for human intake as a nutritional supplementto combat the deficiency of Vitamin B_12_ especially, thereby improving bone strength and stiffness as well as inhibiting the formation of ulcers [60,61]. In addition, numerous studies have showed the medicinal important of *Spirulina* as a rich source vitamin E and β-carotene. Previous investigation by Carcea et al. [62] revealed the generation of vitamin E from *Spirulina* with the content ranging between 2.8 and 12.5 mg/100 g in improved techniques and higher in traditionally dried *Spirulina* [62]. From the experimental samples, th etraditionally dried technique was recorded to be 30.9 mg/100 g content of vitamin E [62]. Research also showed a significant amount of β-carotene in *Spirulina* where the content varied between 33.5 and 231.6 mg/100 g) [62]. High contents of β-carotene were also examined in *Synechoccus* sp. and *Anabaena cylindrica* and in some fruits such as oranges, potatoes, broccoli and carrots [48,63].

Excluding the vascular plants, *Anabaena cylindrica* is also an exceptional source of vitamin C, an antioxidant that could offer a defensive mechanism against oxidants in cyanobacteria [64]. In response to UV radiation stress and preventing destruction of cell membrane, cyanobacteria (*Arthrospira* sp.) could generate a low amount of vitamin D [65,66]. Tarento et al. [67]. emphasized that a significant amount of vitamin K1 could be experimentally examined in the marine plant *Anabaena cylindrica* compared to the content of vitamin K1 in parsley and spinach [67]. 

### 2.3. Peptides

Peptides are small proteins which can be categorized based on their biosynthesis into ribosomal peptides (gene encoded) and non-ribosomal peptides (non-gene encoded). They could also be categorized according to their 3D-structure, covalent binding and modification, source, properties and function [68]. Peptides and polyketides biosynthesis in microbes have exceptional segmental pathways controlled by non-ribosomal peptides (NRPs) and polyketides (PKs). Polyketides (PKs) and non-ribosomal peptides (NRPs) are extensively useful as drugs today, and one possible source for new PKs and NRPs are cyanobacteria. However, there is limited information on the varieties of microorganisms and their PKs and NRPs biosynthetic genes in the marine deposit [69]. NRPS includes components, each of which joins proteinogenic amino acids with non-proteinogenic amino acids, carbohydrates, fatty acids and additional building blocks to form peptide chains [69]. Natural products of PKs and NRPs are secondary metabolites of microbes, used by microbes in adaptation and resistance to environmental stresses [69]. So far, over twenty three thousands natural products of non-ribosomal peptides and polyketides have been recognized and categorized, and they had been extensively applied in medicine as antitumor and antibiotic therapies [70,71].

Biological synthesis of PKs and NRPs are catalyzed by polyketide synthases (PKS) and non-ribosomal peptide synthases (NRPS) [72]. Remarkably, the gene clusters of NRPS and PKS commonly exist in microbes, such as cyanobacteria, compared with eurkarya or archaea [72]. Previous investigations revealed that these gene clusters exist in *Anabaena*, *Nostoc*, *Lyngbya*, *Planktothrix*, *Microcystism*, *Pleurocapsa* and *Nodularia* genera. However, *Nostoc* and *Pleurocapsa* species are known as the common NRP- and PK-producing cyanobacteria [73]. 

Ribosomal peptides (RPs) are proteinogenic amino acids that can be observed on ribosomes that are used as building blocks in ribosomal peptides’ biosynthesis [68,74]. Cyanobacteria also generates lanthipeptides, one of the major ribosomal peptide families including microviridins and cyanobactin. The potential of cyanobacteria to generate lanthipeptides is due to the availability of class II lenthipeptidase and prochlorosins in their genome (such as the marine *Synechococcus* and *Prochlorococcus*) [75,76,77]. 

### 2.4. Terpenoids

Terpenoids are known to be the largest group of bioactive compounds. They are categorized into carotenoids, sesterterpenoids, steroids, hemiterpenoids, diterpenoids, sesquiterpenoids and monoterpenoids. Research shows that over fifty five thousands terpenoids have been discovered so far [78,79]. Triterpenoids were found in a significant amount both in natural cyanobacterium-dominated microbial mats and in laboratory cyanobacterial cultures, for instance, in 2-methylhopanoids (2-MeBHPs) [80,81]. 2-methylhopanoids (2-MeBHPs) belong to pentacyclic triterpenoids, which act as biomarkers for enhanced cyanobacteria in some ecological sites [81]. In addition, 2-MeBHP encourages thawing and freezing resistance, pH stress resistance and osmotic pressure resistance in cyanobacteria to ensure their existence in unfavorable environments such as hot springs, arctic and Antarctic soils, desert soil crusts and in high saline lakes [82,83]. 

Carotenoids, also known as tetraterpenoids, are important for energy dissipation and light-harvesting during the process of photosynthesis [83,84]. Examples of common carotenoids synthesized by cyanobacteria include zeaxanthin, echinenone and β-carotene. Kusama et al. [85] highlighted the importance of zeaxanthin and echinenone to shield PSII (photosystem II) against singlet oxygen [85]. In addition, for protection against direct UV-B radiation, *Pseudanabaena* sp. CCNU1 generates β-carotene, zeaxanthin, echinenone, myxoxanthophyll and canthaxanthin [84]. Carotenoids are necessary to provide protection against photooxidative destruction resulting from direct UV light during the photosynthesis process. Carotenoids are secondary metabolites (lipophilic) generated from the pathways of isoprenoids [84]. Filamentous *Calothrix* PCC 7507 and *Synechocystis* sp. PCC 6803 have been shown to generate sesquiterpenoid geosmin, a sesquiterpene with no isopropyl group [86,87].

Recent studies revealed the biological activities of cyanobacterial carotenoids as a source of medicine for the treatment of several infections. For example, carotenoids and their derivatives obtained from cyanobacteria proved high superoxide anion radical (O_2_^•−^) anti-inflammatory and scavenging effects that facilitate psoriasis treatment [88]. A significant level of carotenoids was observed in *Tychonema* sp. LEGE 07175 and *Cyanobium* sp. LEGE 07175. The extracts revealed a strong antiaging result by hindering hyaluronidase synthesis, the enzyme that promotes hyaluronic acid depolymerization [89]. Antimicrobial sesterterpene from a *Scytonema* sp. (UTEX 1163) culture demonstrated growth inhibition against several pathogenic microbes such as *Mycobacterium tuberculosis*, *Bacillus anthracis*, *Candida albicans*, *Staphylococcus aureus* and *Escherichia coli* [89,90].

## 3. Biotechnological Applications of Cyanobacteria

Cyanobacteria are remarkably the most crucial group of microbes in the universe as they satisfy important environmental roles in the universe, as an important global supplier of oxygen, nitrogen and carbon [91]. Owing to their enormous array of industrial applications, they have been the subject of quite a lot of research. They are important sources of biofertilizers, biofuels, food additives and coloring dyes [92]. Cyanobacteria are used in water treatments, bioplastics production, cosmetics, forestry, feed for animals, production of hydrogen, bioethanol production and biogas production [93,94,95,96,97,98,99]. Here, we review the highly demanded applications of cyanobacteria in biotechnology. 

### 3.1. Cyanobacteria as Food Supplements

Cyanobacterial carotenoids including zeaxanthin, beta-carotene, canthaxanthin and nostoxanthin are great sources used in food supplements, colorants, food additives and animal feed. The productions of these metabolites are on the rise. The supplements are sold in the form of tablets, granules and capsules. As an example, β-carotene, riboflavin, vitamin B_12_ and thiamine are greatly used supplements generated by cyanobacteria such as *Spirulina* [60,66]. In addition, cyanobacteria are known to be used as whole food or as dietary supplements such as minerals, amino acids, proteins, complex sugar, carbohydrate, phycocyanin, active enzymes, essential fatty acids and chlorophyll [100].

*Arthrospira platensis* (a filamentous, gram-negative cyanobacterium) is frequently used as a whole food supplement. It is grown globally and applied as an animal feed supplement in aquariums, poultry, aquariums and many agricultural industries worldwide. Dried *Spirulina* contains 8% fat, 5% water, 51–71% protein and 24% carbohydrate. It is a valuable supplier of several important nutrients and nutritional minerals, including iron and vitamin B_12_. Vitamin B_12_ is important in the production of hemoglobin, maintains the nervous system and participates in DNA synthesis [101,102]. Previous research revealed that several nutritional supplements of cyanobacterial origin such as *Spirulina*, *chlorella* and *Aphanizomenon flos-aquae* are readily available in the consumer markets in the United States [103]. 

Several dietary supplements are frequently generated from the biomass of cyanobacterial species and eaten whole, unlike extracts utilized in pharmaceutical productions [103]. For example, ketocarotenoid (astaxanthin) is seen as a powerful antioxidant compared to vitamin A and vitamin C as well as several carotenoids which perform a crucial role in preventing destruction in human cells via photooxidation. *Haematococcus pluvialis* has been known to generate astaxanthin, which is a strong inhibitor of protease known for the treatment of several diseases including the human immunodeficiency virus disease (the virus that is responsible for the acquired immunodeficiency syndrome (AIDS) which is the final stage of HIV disease) [34,104,105].

### 3.2. Cyanobacteria in Medical and Pharmaceutical Biotechnology

Cyanobacteria is comprise of several secondary metabolites that are useful in the field of medical biotechnology. These microbes have achieved tremendous attention from researchers because of the generation of bioactive compounds that are incredibly useful in medical settings [106]. Although they generate effective toxins, they also generate various metabolites that are vital in terms of their anticancer, antibiotic, anti-inflammatory, immunosuppressant and antimicrobial effects [106,107,108,109,110]. Several global investigations have explored various bioactive compounds from cyanobacteria as anticancer potentials. For instance, Aurilide (isolated from *Dolabella auricularia*) revealed varying cytotoxicity from picomolar (pM) in nanomolar (nM) concentrations against numerous cancer cell lines. Aurilide aids mitochondrial-induced apoptosis by selectively binding to prohibitin 1 (PHB1) in the mitochondria and triggering the proteolytic dispensation of optic atrophy 1 (OPA1) [111,112]. Biselyngbyaside is another drug obtained from *Lyngbya* sp. Biselyngbyaside A shows cytotoxicity against HeLa S3 cells with an IC_50_ value of 0.1 μg/mL. However, Biselyngbyolide B, C, E and F have an antiproliferative impact in HeLa and HL-60 cells, whereas Biselyngbyolide C prompts endoplasmic reticulum (ER) stress and apoptosis in HeLa cells [113,114]. In addition, cryptophycin isolated from *Nostoc* sp. var. ATCC 53789 and GSV 224 is an excellent anticancer agent. Cryptophycin prevents microtubule formation and demonstrates anti-tumorigenic action against various solid tumors implanted in mice involving multidrug-resistant cancer cells. The IC_50_ value of cryptophycin was found to be lower than 50 pM for cell lines multidrug-resistant cancer [115]. In recent studies on cancer therapy, cryptophycin copulated with peptides and antibodies had been developed for targeted drug delivery [116]. The antiproliferative actions of Arg-Gly-Asp (RGD)–cryptophycin and *iso*Asp−Gly−Arg (*iso*DGR)-cryptophycin conjugates were experimented against human melanoma cell lines (M21 and M21-L). The investigation revealed that the conjugations exhibit anticancer efficacy at nanomolar concentrations with diverse expression integrin α_v_β_3_ (a type of integrin that is a receptor for vitronectin) levels [116,117]. 

Apratoxin and its derivatives, developed from several types of marine cyanobacteria (such as *Moorea producens* strain (RS05), *Lyngbya bouillonii*, *L. sordida*, *L. majuscula*, genera *Phormidium*, and genera *Neolyngbya*), are known to combat diverse forms of cancer cell lines [106,118,119]. Curacin A islolated from Lyngbya *majuscula* is effective against cancer of the breast [120]. Consequently, in addition to the natural resources, cyanobacteria provide a favorable means, presenting a comprehensive variety of substances for new drug discovery and development [121]. Secondary metabolites of cyanobacteria can be applied as natural compositions in cosmetology [122]. For instance, they can be used as photoprotective Mycosporine-like Amino Acids (MAAs) in sunscreens to shield the skin from destructive UVR. In addition, cyanobacteria natural pigments such as phycobiliproteins and carotenoids may be employed as natural colorants as well as antioxidants to shield the skin from destruction resulting from exposure to UV irradiation [123]. Therefore, the field of cyanobacteria exploration is very crucial research.

### 3.3. Cyanobacteria in Bioplastic and Biofuel Production

Cyanobacteria features such remarkable approaches for fixation and absorption of atmospheric nitrogen and CO_2_, and employing them it for growth in unfavorable climatic environments, such as unfertile soils and saline waters, make them extremely suitable for production of biodegradable plastics and biofuels [124]. Although numerous means are available for bioplastic and biofuel production, cyanobacteria have been studied as energy-rich suppliers due to the triacylglycerol (TAG) and diacylglycerol (DAG) production, which can be employed as biodiesel precursors [125]. Cyanobacteria such as *Synechocystis*, *Spirulina*, *Anabaena* and *Nostoc muscorum* can function as bio-factories for biofuel and bioplastic generation. For instance, they have the metabolism for generating cost-effective and sustainable biopolymer polyhydroxyalkanoates (PHAs), and polyhydroxybutyrate (PHB), among other copolymers [34]. Biopolymeric PHB presents material features such as polypropylene, a standard plastic obtained from petroleum (fossil fuels). However, in comparison to standard plastics, PHB is biodegradable, and its application as a complementary of standard plastics can assist in alleviating the critical ecological influences of fossil fuels and plastics in nonbiodegradable overconsumption [126,127]. 

Species of cyanobacteria such as *Synechococcus* and *Synechocystis* species can generate lactate and succinate which are important chemicals in the production of bioplastics. Succinate is an essential biotechnological chemical, a precursor of adipic acid, 1,4-butanediol, and other four-carbon chemicals [128]. In recent study by Durall et al. [128], the highest succinate titer was achieved in dark incubation (compared to the light and anoxic darkness conditions) of an engineered cyanobacteria strain (*Synechocystis* PCC 6803), coupled with a limited glyoxylate shunt (*aceA* and *aceB*) overexpressing isocitrate lyase with phosphoenolpyruvate carboxylase, with supplemented medium using 2-thenoyltrifluoroacetone [128]. Furthermore, a research team at Kobe University [129] illustrated the method by which *Synechocystis* sp. PCC 6803 (one of the most global researched cyanobacterial strains known to be the model microbe for photosynthate production because of its fast growth and the ease of genetic manipulation) could generate D-lactate (utilized in biodegradable plastic productions). The research team demonstrated that malic enzyme accelerates the production of D-lactate via genetically modifying D-lactate synthesis pathways using cyanobacteria. They eventually succeeded in producing the world’s maximum amount of D-lactate (26.6 g/L) directly from light and carbon dioxide [129,130]. 

The process of generating petroleum products (such as jet fuel, heating oil, propane, gasoline, and diesel) could be a serious potential environmental threat (pollution), and the release of toxic substances such as greenhouse gasses (methane, ozone, carbon dioxide (CO_2_) and nitrous oxide) into the atmosphere could be hazardous to humans, plants and animals [131,132]. Cyanobacteria provide excellent assurance as suppliers of renewable biofuels for the energy sector [133]. Biofuels such as 2-methyl-1-butanol, isobutanol, 2,3-butanediol ethanol, isobutanol and ethylene have been produced from engineered cyanobacteria (*Synechocystis* sp.) [134]. Biofuels including gasoline, jet fuel, biodiesel and ethanol are being generated from genetically modified cyanobacteria by some US-based companies such as Joule Unlimited and Algenol [15,135,136,137].

### 3.4. Cyanobacteria in Bioremediation

Cyanobacteria are characterized by high adaptability to various stress conditions, being at the same time quite resistant to toxic compounds of different origins [31]. Therefore, such photosynthetic bacteria seem to be relevant for numerous approaches in the field of bioremediation, including soil remediation, wastewater treatment and degradation of organic pollutants. Water pollution represents a real environmental problem as a consequence of anthropogenic activity both in the context of urbanization and industrialization of the environment as well agricultural practices. Several research groups have successfully explored the potential application of cyanobacteria for wastewater treatment, demonstrating that polluted water from different sources can be treated effectively with the help of such microorganisms, and the system based on photosynthetic procaryotic cells can be considered as a promising alternative to conventional biological processes such as activated sludge. 

Extensive aquaculture generates a huge amount of polluted nitrogen-rich water released into the costal seas, directly improving the nitrogen pool in marine ecosystems, altering the balance of species [138]. The marine cyanobacterium *Synechococcus* sp. has been shown to effectively remove ammonium from brackish aquaculture wastewater [139]. The tested microorganism assimilated ammonium through the actions of glutamine synthetase (GS, EC 6.3.1.2) and glutamate synthase (GOGAT, EC 1.4.1.13) cooperated in the GS-GOGAT cycle, which is closely related to the adaptive strategy of the *Synechococcus* species to changing nutrient conditions [140].

An interesting approach to biological wastewater treatment is the use of a cyanobacterial-bacterial consortium that operates on the synergistic action between photosynthetic microorganisms and heterotrophic bacteria. It should be stressed that such photosynthetic microbes are known from exopolysaccharides production that are useful in establishing a symbiotic association of cyanobacteria with other organisms [141]. Brewery wastewater containing high concentrations of organic pollutants and significant amounts of nitrogen and phosphorus has been treated using a cyanobacterial-bacterial consortium dominated by the filamentous cyanobacterium *Leptolyngbya* sp. [142] Cyanobacterial-bacterial aggregates grown under optimal pH and temperature conditions were found to be highly effective and successfully reduced the levels of nitrogenous compounds, including nitrate (up to 80%), ammonium (up to 90%) and phosphorus compounds (up to 70%) in crude wastewater. The introduction of an additional wastewater pretreatment step involving electrocoagulation followed by the use of an electrochemically treated supernatant as a medium for the cultivation of microorganisms increased the level of removal of pollutants [143]. The bioremediation potential of the studied consortium was successfully verified under stressed conditions in a flat-plate photobioreactor filled with hydrophilic support [144]. The proposed solution is very important from a technological point of view as it brings laboratory ideas closer to the practical disposal of brewery wastewater. Other examples illustrating the applicability of bacterial consortia for wastewater treatment are as follows: *Dinophysis acuminata* and *Dinophysis caudata* living in a consortium were reported to effectively remove phosphate, phenol and cyanide from coke-oven wastewater [145], the effective mixing of nitrogen-fixing soil cyanobacterial culture was municipal wastewater treatment [146] and mixed cyanobacteria formed mats degraded pesticide lindane in pesticide-contaminated effluents, showing high resistance against its toxicity [147].

The most important demonstration of the degradation potential of cyanobacterial cells is their ability to remove heavy metals from sewage of various origins. The ability of cyanobacteria to tolerate and interact with metal ions makes them an attractive tool for environmental biotechnology [148]. Cyanobacteria employ a variety of mechanisms such as biosorption, bioaccumulation, activation of metal transporters, biotransformation, and induction of detoxifying enzymes to sequester and minimize the toxic effects of heavy metals [149]. Hexavalent chromium can be present in some aquatic systems as a result of textile, paint, metal cleaning, plating, electroplating, and mining industries [150]. As chromium (Cr(VI)) is potentially toxic and carcinogenic to humans, its removal from water and wastewater is required to avoid serious health and environmental problems. The living cyanobacterial consortium consisting of *Limnococcus limneticus* and *Leptolyngbya subtilis* has been found to be efficient in the removal of Cr (VI) from wastewater [151], whereas the mat-forming cyanobacterial consortium consisting of *Chlorella* sp., *Phormidium* sp. and *Oscillatoria* sp. efficiently removed hexavalent chromium from the sewage in the tannery industry [152]. It is worth noting that cultivation of microbes under conditions that force cells to organize themselves into a mat or application of naturally forming mats can be applied on a large scale and used practically in bioremediation. Cadmium is a toxic metal and its exposure remains a global concern [153]. An interesting strategy for the sequestration of Cd (II) from aqueous solutions was proposed, including axenic cultures of *Nostoc muscorum* immobilized on the glass surface through the formation of biofilms [154]. A microorganism growing as a biofilm expressed the ability to adsorb Cd(II) in a wide concentration range of ~24 ppb to 100 ppm and a pH range of 5 to 9. Strains belonging to *Nostoc* genera that produce EPS are known for their ability to remove metals, and the application of mixotrophic cultivation conditions increased the uptake capacity of heavy metal ions by the *Nostoc* species [155]. The self-flocculating *Oscillatoria* sp. was shown to possess metabolic properties to eliminate Cd from metal-contaminated water [156]. Authors analyzed the mechanism of bacterial activity against Cd and they found that metal adsorption by negatively charged functional groups in cyanobacterial biomasses was the major mechanism used by *Oscillatoria* sp. to remove metals from the aqueous medium followed by Cd bioaccumulation in living cells. The potential in municipal sewage remediation has also been shown for *Anabaena oryzae,* characterized by a high removal efficiency for cadmium, lead, zinc, iron, copper and manganese [157].

Cyanobacteria are present very often in polluted environments [158], and due to their naturally evolved resistance and selectivity against environmental pollutants, they exude a significant metabolic potential for xenobiotic degradation. For example, *Spirulina* spp. demonstrated the ability to metabolize the phosphonate xenobiotic Dequest 2054^®^ [159], *Leptolyngbya* sp. has the ability to degrade phenol, significantly decreasing its concentration in the cultivation medium [160], *Aphanothece conferta* demonstrated high degradation efficiency against aliphatic hydrocarbons, whereas aromatic hydrocarbons were degraded by *Synechocystis aquatilis* [161]; cyanobacteria have also been reported to have the relevant enzymatic system to participate in the degradation of textile dyes [162,163]. 

### 3.5. Applications of Cyanobacteria Species in Biocatalytic Processes

Cyanobacteria, as organisms capable of photosynthesis, are a group of biocatalysts with unusual activities that can find application in biocatalytic processes for obtaining crucial derivatives used in various industries. One of the derivatives with extensive use in the pharmaceutical and fragrance industry are unsaturated alcohols, e.g., cinnamyl alcohol, essential both for the synthesis of Taxol and Chloramycin and for the production of aromatic compounds [164,165]. Traditionally, this compound is obtained from cinnamaldehyde of natural origin, but the chemical reduction of the aldehyde group to a primary hydroxyl group with the double bond intact is problematic, so a mixture of products (Figure 1) is usually obtained. The solution to this problem may be the use of selected cyanobacterial strains that can chemoselectively reduce aldehyde to alcohol without breaking the double bond in the side chain [165]. 

Yamanaka et al. [165] have selected the strain *Synechocystis* sp. PCC 6803 as the most efficient, capable of converting cinnamaldehyde to cinnamyl alcohol with an efficiency of up to 98%. In addition, they demonstrated that the reaction is strictly light-dependent, and the coenzymes (NADPH) needed for reduction are regenerated in photosynthetic electron transfer reactions; thus, there is no need for additional supporting substrates, which greatly simplifies the whole process. 

Another group of compounds of industrial importance are optically pure secondary alcohols used as chiral building blocks for the synthesis of optically active products. The whole cell biocatalytic reduction of the corresponding ketones is a favorable alternative to traditional chemical processes. Acetophenone is often used as a model substrate for screening in case of bioreduction and it can undergo enantioselective bioreduction to *R*- or *S-*phenyl ethanol (**1**, Figure 2). Several methods have been developed using whole cells of heterotrophic microorganisms or plant tissues [166,167,168], but typically these processes require co-substrates to support coenzyme regeneration systems. The intracellular oxidoreductase that is responsible for the asymmetric acetophenone reduction reaction is not itself dependent on light but requires the presence of reduced NAD(P)H, which is one of the products of the light phase of photosynthesis (Figure 2). Lighting is therefore essential for the regeneration of the cofactor. According to the literature data, cyanobacterial enzymatic systems could be effectively applied for the reduction of acetophenone to *S*- phenyl ethanol (e.g., *Spirulina platensis*–45% yield, 97% e.e [169]; *Arthrospira maxima*- 45,8% yield, 98,8% e.e. [170]; however, the efficiency and enantioselectivity of processes are dependent on the growth rate of biocatalysts, substrate concentration and light regime [169,170].

Similar relationships have been observed in the reduction of diethyl 2-oxophosphonates to corresponding *S*-2-hydroxyalkylphosphonates (Figure 2), which in an optically pure form can also be used as a synthon for the synthesis of derivatives used in medicine to agriculture [171,172,173]. The biological activity of phosphonates is based, among other things, on enzyme inhibition and therefore they are difficult substrates for biocatalytic processes, especially in the scaling-up context. Additionally, phosphonates with a carbonyl group that is located right next to the aromatic ring were inefficiently transformed by fungal biocatalysts [174,175]; therefore, special attention was turned to the strain of *Nodularia sphaerocarpa* capable of reducing the substrate (**3**) (Figure 2), with a yield of 99% and an enantiomeric excess of 93%. As in the case of acetophenone, the efficiency of the reaction was closely correlated with the concentration of the substrate used and the best results were obtained using 1 mM of diethyl 2-oxo-2-phenylethylphosphonate (**3**) [176]. Other substrates tested in the discussed study were not reduced very effectively (diethyl 2-oxopropylphosphonate (**2**)—transformed with *Arthrospira maxima* with 20% yield and 99% e.e, diethyl 2-oxobutylphosphonate (**3**)—transformed with *Nodularia sphaerocarpa* with 27% yield and 80% e.e.) [176]. In the next stage of the research mentioned above, immobilization in calcium alginate was used to increase the resistance of the biocatalyst to the toxic effects of substrates, and the influence of shaking (better contact of the substrate with the biocatalyst) on the reduction efficiency was checked. Also, in this case *Nodularia sphaerocarpa* turned out to be the most effective strain. The use of mixing or immobilization made it possible to increase the scale of the process. Packing the immobilized photobiocatalyst in a simplified flow reactor allowed for increasing the substrate (**3**) (2-oxo-2-phenylethylphosphonate) concentration to the value of 10 mM, although in the 500 mL batch culture conducted with stirring, a better efficiency (44% of yield, 91% of e.e.) was obtained compared to 38% of the yield and 86% of e.e. obtained in the column reactor [177]. 

A quite recent finding was that NAD(P)H or flavin-dependent enzymes involved in many light-independent metabolic processes may have unnatural activities after exposure to light. Cofactors of these enzymes can form electron-donor-acceptor (EDA) complexes with unnatural substrates (Figure 2). EDA can be excited by visible light which allows for the flow of electrons and consequently the formation of the reduced product [178]. The first enzyme of this type to be described was NADPH-dependent carbonyl ketoreductase, which catalyzed the radical dehalogenation of halolactone upon exposure to light [179]. Later, in a similar manner, asymmetric reductive cyclization [180,181], intermolecular hydroalkylation [178] or asymmetric hydrogenation [182] using ene-reductase was achieved.

Another meaningful application of cyanobacteria is related to the activity of the fatty acid photodecarboxylase (FAP) of microalgae *Chlorella variabilis* origin. This enzyme is inside the cell, and is involved in the lipids metabolism and driven by blue light and the presence of FAD. This mechanism was deeply analysed and described by Damine Sourigue et al. [183]. Scientists studied the microalgal strain *Chlorella variabilis* NC64A and discovered that phototrophs produced by photodecarboxylase belong to the oxidoreductases and catalyses decarboxylation of saturated and unsaturated fatty acids, with the releasing of the corresponding alkanes or alkenes. This enzyme interacts with substrates by binding them in a tunnel-like site, which leads directly to the flavin dinucleotide, which is crucial as a moiety sensitive to light excitement, which is followed by the electron transfer from the substrate. Such a sequence initiates the reaction. This discovery has not gone unnoticed by researchers involved in the application of biocatalysts of different origins for the synthesis of variable chemical compounds according to green chemistry rules. Photodecarboxylase from *Chlorella variabilis* NC64A (Cv FAP) was considered a part of the cascade of reactions finally leading to biofuel production. The very first approaches were focused on the evaluation of the activity of the enzyme (Cv FAP) towards structurally different fatty acids (saturated and unsaturated) [184,185]. Usually, reactions were carried out at pH 8.5 under illuminations by blue light of intensity = 13.7 μEL^−1^:s^−1^ and the proportions of the substrate and enzyme were as follows: [substrate] = 30 mM, [decarboxylase] = 6.0 μM. According to the general scheme (Figure 3) below, a number of fatty acids were tested: lauric, myristic, palmitic, margaric, stearic, oleic, linoleic and arachidic. 

The best conversion degrees were obtained for the four substrates: palmitic, margaric, stearic and arachidic acids (above 90%). The differences in the results were correlated to the differences in the substrates biding, which in turn is due to differences in matching between the tunnel site in enzyme structures and the substrate moiety. However, further experiments were meant for the scaling of the selected reactions and applying the palmitic acid as a starting compound; this approach succeeds and leads to the scale elevation up to 155 mg of pentadecane production. These results were important in relation to the necessity of green chemistry solution implementations in the chemical industry. The practical side of the mentioned results appeared in the cascade of the reactions, starting from triglycerides hydrolysis via decarboxylation and leading to the release of glycerol, hydrocarbon and carbon dioxide (Figure 4) [184,185,186]. Previous approaches were based on the two-step process design as a sequence starting from the hydrolysis conducted by the lipase from *Candida rugosa,* which delivers fatty acids for the next reaction catalysed by photocarboxylase (Figure 4). 

Further developed protocols were performed as one-step ones. In addition, the mode of the biocatalysts applied for photodecarboxylations was modified. Instead of using the purified enzyme, whole-cell biocatalysts were involved in the reaction. Genetically engineered cells of *E. coli* were able to produce photodecarboxylase at ae high level and with great activity towards studied substrates, which is why for the next set of experiments such a biocatalyst was applied. The limiting factor was finding a compatible reaction–environment lipase, which is at the same time insensitive to blue light, which is essential for decarboxylase activity. These boundary conditions were met by immobilized lipase from the fungus *Rhizopus oryzae* [186] and finally the one pot sequence of reactions was set. The effectiveness of this protocol was checked against plants oils (e.g., soybean oil) and waste cooking oil and for different physical chemical parameters (e.g., temperature, biocatalysts and substrates concentrations, reaction duration). This allowed for selecting the best protocol for scaling (from 1 mL to 15 mL of the final volume of reaction mixture), which was conducted with soybean oil with the receiving of almost 1 g of hydrocarbon (21.2% of isolated yield). 

The above achievements are the base for further discoveries as they proved the practical and biotechnological function of photobiocatalysts; nevertheless, it should be emphasized that they require the individual protocols for almost every designed biocatalytic process.

## 4. Conclusions and Future Perspective

This review has highlighted that cyanobacteria are promising sources of several primary and secondary metabolites that are known for their biological activities. These bioactive compounds were reviewed to possess several impacts in various biotechnological industries. Nonetheless, to improve the usefulness of cyanobacteria for human and animals exploitation, extensive research is essential for profiling cyanobacterial secondary metabolites, known to be generally employed in different countries in the areas of nutraceuticals and pharmaceuticals applications, including in antimicrobial and antitumor activities. Furthermore, cyanobacterial enzymatic systems are currently applied as biocatalysts for conducting chemically different reactions to achieve various synthetic goals. This field is still not fully explored and offers biological tools of different features compared with light independent organisms (e.g., fungi and heterotrophic bacteria). There is a need for further research on how to effectively exploit cyanobacteria using green technological advancements as well as efficient insights on cyanobacteria applications on larger scales. Future investigation that is also promising is essential to explore more beneficial roles of cyanobacteria in biotechnology.

## Data Availability

Study does not report any data.

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
