# Peer review of "Versatile Applications of Cyanobacteria in Biotechnology"

_microorganisms, 2022, doi:10.3390/microorganisms10122318_

Round 1
Reviewer 1 Report
The study covered a comprehensive introduction towards the five applications of cyanobacteria: from nutrition source to bio-analytic tool. However, each part only reviewed the basic and well known knowledge and in-depth analysis, discussion and future prospective is lacked for each section. I suggest the authors to reorganize study: only cover one or two applications with clearer logic structure and in-depth discussion.
Author Response
Thank you for the review and comments. We have added the conclusion and future perspective to the review. However, this review includes representative examples of diverse applications of cyanobacteria in biotechnology and it covers the scopes which the authors intended this review to be structured. Also, because of the nature of this article, which is composed basing on the current literature data, the disscussion is only at the essential level pointing the key trends in algal biotechnology. Article is suported by a number of references, which include the deeper analysis of particular issues.
Reviewer 2 Report
The review is well written and presented especially in the first part. However, when the "Applications of cyanobacteria species in biocatalytic processes" is presented, I found this part not totally focused on the title of the paper. If we are speaking of "versatile" application of cyanobacteria I expect not only the biocatalitic process. Cyanobacteria is well established as source for food, energy, raw material, EPS, biofertilizer etc etc. In all these field we have useful biotechnology approach. What I suggest is to add some of these other applications to obtain a review with a complete point of view.
Author Response
Thank you for the review and comments. We had gone through the review, and what we portray is by letting the readers and the general public to be aware of the versatility of how cyanobacteria is applied in several spheres of biotechnology. “Applications of cyanobacteria species in biocatalytic processes” is only a segment of the whole article that wrapped-up it’s laboratory applications as biocatalysts.
Round 2
Reviewer 1 Report
The revision (last section) improved the manuscript.
Reviewer 2 Report
The addition of last part is well thought and allow to better understand the aim of review